# Glutamine and Albumin Levels in Cerebrospinal Fluid Are Correlated with Neurological Severity in an Experimental Model of Acute Hepatic Encephalopathy

**DOI:** 10.3390/metabo15090598

**Published:** 2025-09-08

**Authors:** Pedro Arend Guazzelli, Felipe dos Santos Fachim, Anderson Santos Travassos, Caroline Casagrande Schaukoski, Pâmela Cristina Lukasewicz Ferreira, Fernanda Uruth Fontella, Marco Antônio de Bastiani, Adriano Martimbianco de Assis, Diogo Onofre Souza

**Affiliations:** 1Graduate Program in Biological Sciences: Biochemistry, ICBS, Universidade Federal do Rio Grande do Sul—UFRGS, Porto Alegre 90035-003, Brazil; paguazzelli@gmail.com (P.A.G.); anderson.travassos@gmail.com (A.S.T.); furruth@gmail.com (F.U.F.); 2Graduate Program in Pharmaceutical Sciences, Universidade Federal do Rio Grande do Sul, Av. Ipiranga 2752, Porto Alegre 90610-000, Brazil; pamlukasewicz@gmail.com; 3Graduate Program in Biological Sciences: Pharmacology and Therapeutics, Universidade Federal do Rio Grande do Sul, Porto Alegre 90050-170, Brazil; tyrev@hotmail.com; 4Graduate Program in Health and Behaviour, Center of Health Science, Universidade Católica de Pelotas—UCPel, Pelotas 96015-560, Brazil; 5Department of Biochemistry, Universidade Federal do Rio Grande do Sul—UFRGS, Porto Alegre 90035-003, Brazil

**Keywords:** hepatic encephalopathy, acute liver failure, neurological severity grade, CSF glutamine, CSF albumin

## Abstract

Background/Objectives: Hepatic encephalopathy (HE) is a severe neurological complication of acute liver failure (ALF) characterized by the accumulation of neurotoxic metabolites and impaired cerebral function. We aimed to examine the correlation between HE severity and cerebrospinal fluid (CSF) biomarker levels in a rat model of ALF induced by subtotal hepatectomy. Methods: Male Wistar rats underwent 92% hepatectomy and were monitored for neurological impairment via a standardized HE score. At twenty-four hours post surgery, CSF and blood were collected for biochemical analysis. Results: We found a significant positive correlation between neurological severity and CSF levels of glutamine (r = 0.929, *p* < 0.001) and albumin (r = 0.869, *p* < 0.001), both with HE grade I scores, highlighting their prominent role as HE biomarkers. Other amino acids, including aspartate (r = 0.790, *p* < 0.001), glutamate (r = 0.853, *p* < 0.001), isoleucine (r = 0.834, *p* < 0.001), leucine (r = 0.813, *p* < 0.001), lysine (r = 0.861, *p* < 0.001), methionine (r = 0.889, *p* < 0.001), phenylalanine (r = 0.916, *p* < 0.001), ornithine (r = 0.775, *p* < 0.001), tryptophan (r = 0.814, *p* < 0.001), and valine (r = 0.721, *p* < 0.001), also showed significant correlations with HE severity but not with HE grade I scores. Conclusions: These findings underscore the potential of glutamine and albumin in CSF as key biomarkers for assessing neurological severity in ALF patients.

## 1. Introduction

Acute liver failure (ALF) refers to the rapid impairment of hepatic function. ALF patients are at high risk of developing life-threatening conditions, including hepatic encephalopathy (HE), infections, coagulation disorders, and multiorgan failure, and require specialized medical support in intensive care units [1,2].

The clinical management of ALF presents a significant challenge to health care services, often necessitating drastic interventions such as liver transplantation (LT) [3,4,5]. Various clinical severity criteria, such as the King’s College criteria, the Clichy criteria, and the MELD score, are used to identify patients who would benefit from LT [6,7,8].

HE, defined as brain dysfunction caused by ALF, is the major cause of death in ALF patients [3,9,10]. Patients with mild HE (grades I and II) have a high likelihood of spontaneous recovery (approximately 70%), whereas those with advanced HE (grades III and IV) have a much poorer prognosis (approximately 20% recovery rate) [7,10]. As the prognosis of ALF patients is linked to HE severity, identifying biomarkers of brain impairment is crucial for assessing disease progression [11,12,13].

Animal models of liver disease have been used to understand HE pathophysiology and explore new strategies for mitigating disease severity [14] based on various HE severity scales. The most robust rat studies, dating back more than 15 years, used portocaval shunting, hepatic artery ligation, and galactosamine-induced ALF to induce HE [15,16,17].

As the link between ALF prognosis and HE severity is a very relevant theme [11,12,13], numerous experimental studies, including those from our group [18], have sought to identify biomarkers for assessing the degree of neurological HE severity in various ALF animal models [15,19]. These studies revealed sustained high serum and CSF levels of glutamine [20], ammonia [21], and albumin [22], which are similar to findings from clinical studies [20,23,24,25,26]. Recent data suggest that high serum glutamine levels may indicate the severity and patient outcomes in ALF [20]. Furthermore, studies showing increased CSF and serum levels of biomarkers of brain impairment indicate that ALF-induced alterations may affect blood–brain barrier (BBB) function [27,28,29,30,31].

Therefore, the aim of this study was to identify a positive correlation between HE severity and the levels of glutamine, albumin, and several amino acids in CSF in an animal model of ALF. This correlation could serve as a valuable prognostic indicator for HE severity.

## 2. Methods

### 2.1. Animals and Experimental Design

Ninety-day-old male Wistar rats from the Animal House of the Department of Biochemistry, ICBS, Federal University of Rio Grande do Sul, Porto Alegre, Brazil, were used. They were housed 4 per cage in a colony room maintained on a 12 h light/dark cycle (lights on 7:00–19:00) at 22 ± 1 °C, with ad libitum access to water and standard commercial chow (SUPRA, Porto Alegre, Brazil). The experimental protocol (project number 29468) was approved by the Ethics Committee for Animal Research of the Federal University of Rio Grande do Sul, Porto Alegre, Brazil, and adhered to the NIH Guide for the Care and Use of Laboratory Animals (NIH publication 85–23, revised 1996). All efforts were made to minimize animal numbers and suffering. Any animal that exhibited intolerable pain, hemorrhage, or seizures during surgery (sham or HE) was immediately euthanized by exsanguination; however, no rats presented these complications. Three rats died unexpectedly during surgery (sham: n = 1; hepatectomy: n = 2).

The incidence, symptomatology, natural course, and clinical outcomes of liver diseases vary between sexes [32], with men being 2-fold more likely to die from chronic liver disease and cirrhosis than women [33]. This study was therefore performed in male rats.

### 2.2. Timeline of the Experimental Design

#### Cohorts of Animals (Figure 1)

Cohort 1: Evaluation of HE severity every 6 h after surgery for a total of 72 h (sham: n = 9 and hepatectomy: n = 32) by using the adapted HE neurological severity grade (Figure 1).

**Figure 1 metabolites-15-00598-f001:**
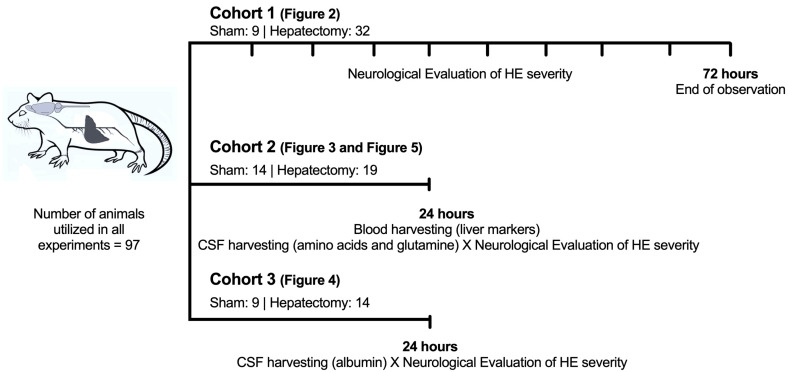
Timeline of the experimental design. The animals were divided into 3 different cohorts (n = 97). Cohort 1: Adapted neurological HE severity scale for evaluating rats up to 72 h after hepatectomy (n = 41). Cohort 2: Neurological HE severity scale, blood liver function biomarkers, and amino acid levels in CSF 24 h after surgery (n = 33). Cohort 3: Neurological HE severity scale and albumin levels in CSF 24 h after surgery (n = 23).

Cohort 2 (24 h after surgery): Evaluation of plasma levels of liver functionality biomarkers (sham: n = 14 and hepatectomy: n = 19), and correlations of CSF albumin levels with adapted HE neurological severity grades—sham (n = 9), grade I (n = 6), grade II (n = 5), grade III (n = 3), and grade IV (n = 0).

Cohort 3 (24 h after surgery): Correlations of glutamine and amino acid levels in CSF with adapted HE severity grades (sham: n = 14 and hepatectomy: n = 19).

### 2.3. Surgical Procedure for Triggering HE

All surgical procedures were performed randomly. The subtotal hepatectomy model has been used as an experimental model of HE in [18,34,35,36,37], and it was used in the present study with reference to [35]. Anesthesia was induced and maintained with 3% isoflurane at an oxygen flow rate of 1 L/min throughout the procedure. After the abdominal wall was cleaned with povidone-iodine, a median laparotomy was performed, and the liver ligaments and adhesions were cut. The hepatic ligaments were resected, and the pedicles of the anterior lobes were ligated with a 4–0 silk thread to interrupt the blood flow and allow lobe resection. The same procedure was performed on the right lobes. Only the omental lobes (8% of the liver mass) remained functional [38]. The post-surgical mortality rate (Figure 2) of rats subjected to partial hepatectomy (n = 32) up to 72 h was 90.6% (n = 29).

To reduce pain, lidocaine (0.8 mL) was administered intramuscularly at the wound borders after the abdominal wall was sutured. The rats were subsequently placed in a heated box (25 °C) for 30 min before being returned to their home cages. The animals had free access to 20% glucose in their drinking water throughout the experiment [39,40]. Additionally, 3 injections of the same glucose solution were administered (2 mL/kg, i.p.) at 0, 6, and 12 h after surgery to avoid hypoglycemia. Sham surgery was performed following the same protocol, except for pedicle ligation and liver resection.

Our group used this surgical ALF model in recent studies [18,35] to obtain data regarding brain energy metabolism and electroencephalographic alterations [18,35].

### 2.4. HE Neurological Severity Scale (Table 1)

The grading scale was adapted from the literature [15,16,41], aiming to improve the evaluation of the severity scale. Neurological evaluations were conducted blindly. HE neurological severity evaluations were performed every 6 h after the surgical procedure for 72 h (Figure 2). Initially, the cage grid was removed to stimulate animal locomotion (time 0). Animals exhibiting some spontaneous locomotion or exploratory activity (e.g., interacting with other animals, food, or cage walls) within the first 10 s were considered grade 0 HE in terms of neurological severity. Animals that remained immobile were tested for righting and corneal reflexes. The righting reflex was assessed by lifting the animal and trying to place it on its back on the cage floor. A normal response refers to the animal turning its body to land on its feet before reaching the ground or immediately returning to its normal position once released by the researcher. The corneal reflex was tested by softly poking the cornea with a cotton swab to trigger an eyelid closure response. Animals that did not move within the first 10 s but exhibited no abnormal reflexes were considered grade I HE in terms of neurological severity. Animals were considered grade II HE in terms of neurological severity if they showed no spontaneous locomotion and allowed themselves to be positioned on their backs but then quickly (<3 s) returned to their normal position. Animals were considered grade III if they exhibited significant locomotor deficits with no spontaneous locomotion or exploration and were unable to return to their upright position when placed on their back but presented normal corneal reflexes. Animals with an abnormal righting reflex or corneal reflex or no response to pain stimuli were considered grade IV HE in terms of neurological severity. The time of death (grade V of HE neurological severity) was registered as the last time the animal was seen alive. The sham group presented only grade 0.

### 2.5. Measurement of Plasma Levels of Liver Functionality Biomarkers

Twenty-four hours after surgery, the animals were briefly anesthetized with 3% isoflurane, and blood was collected via direct cardiac puncture with a sodium citrate tube and centrifuged at 5000× *g* for 10 min. The plasma was stored at −80 °C for measurement of the levels of biomarkers aspartate aminotransferase (AST), alanine aminotransferase (ALT), total and direct bilirubin, and ammonia, as well as prothrombin time, utilizing commercial kits (Labtest Diagnostica S.A., Lagoa Santa, Brazil).

### 2.6. Measurement of Cerebrospinal Fluid (CSF) Harvest (24 h After Surgery)

CSF collection was performed according to previous reports from our group [18,35,42]. Twenty-four hours after surgery, the animals were anesthetized with 3% isoflurane and positioned in a stereotaxic frame. CSF samples (approximately 50 μL) were collected via direct puncture of the 4th ventricle (cisterna magna) utilizing 29-gauge insulin syringes. The collected CSF was immediately centrifuged at 1000× *g* for 10 min, and the supernatant was stored at −80 °C for subsequent analysis. On the day of measurement, the CSF samples were filtered through a 0.22 μm filter, further deproteinized with methanol (1:5 water), and centrifuged at 3000× *g* for 10 min, after which the supernatant was collected. The evaluation of amino acids in CSF was performed through HPLC in partnership with Laboratory Genetics Unit, Medical Genetics Service, Clinicas Hospital of Porto Alegre—HCPA, (Porto Alegre, Brazil), when it was evaluated: (i) Limit of detection/quantification (LOD/LOQ, 2.0 μM); (ii) Precision: intra-day (6.0%) and inter-day (8.5%); and (iii) Recovery of standards (ranged between 82% and 120%), accordingly to Joseph [43].

### 2.7. Measurement of CSF Albumin Levels (24 h After Surgery)

Twenty-four hours after surgery, the levels of albumin in the CSF were determined via high-performance liquid chromatography coupled with a fluorescence detector (HPLC-FLD) consisting of an LC Shimadzu system (Shimadzu, Kyoto, Japan) equipped with an LC-20AT pump, a DGU-14A degasser, a thermostat set for a CTO-10A column, and a fluorescence detector (RF 20A). LC Solution software was used for data acquisition and processing. The FLD was set at 278 nm (excitation) and 335 nm (emission). An Agilent ZORBAX SB-C18 reversed-phase column (5 μm particle size, 250 × 4.6 mm i.d.) was used. We used gradient conditions consisting of solvent A (H2O + 0.1% formic acid) and solvent B (acetonitrile [ACN]) as follows: A → 65% B → 35% (0–5.0 min), A → 70% B → 30% (5.0–10 min), and A → 65% B → 35% (10.0–17.0 min). The flow rate was 0.7 mL/min. FDA bioanalytical guidance, complemented by EMA guidance, was used for method validation. Albumin stock solutions were prepared in water at a concentration of 1 mg/mL and stored at −20 ± 2 °C. For each day of analysis, standard solutions of albumin were prepared at 0.1, 0.5, 1, 10, 50, and 100 µg/mL. Sample preparation was performed by adding 10 µL of CSF to 40 µL of ACN and vortexing. The solution was transferred to conical vials, and 10 µL was injected into the HPLC [44].

### 2.8. Measurement of Amino Acid and Glutamine Levels in CSF (24 h After Surgery)

Twenty-four hours after surgery, the levels of amino acids and glutamine in CSF (alanine, aspartate, glutamate, isoleucine, leucine, lysine, methionine, ornithine, phenylalanine, tryptophan, valine, and glutamine) were determined via high-performance liquid chromatography (HPLC) with Shimadzu Instruments [18]. For the analysis, a reverse-phase column (Supelcosil LC-18, 250 mm × 4.6 mm × 5 μm, Supelco) (50 µL loop injection valve, 40 µL injection volume) was used, and fluorescence detection was performed after precolumn derivatization using 100.5 µL of OPA (5.4 mg of OPA in 1 mL of 0.2 M sodium borate, pH 9.5) plus 25.5 µL of 4% mercaptoethanol for 3 min. The mobile phase flowed at a rate of 1.4 mL/min, and the column temperature was 25 °C. The buffer composition was as follows: buffer A, 0.04 mol/L of sodium dihydrogen phosphate monohydrate, pH 5.5, containing 80% methanol; buffer B, 0.01 mol/L of sodium dihydrogen phosphate monohydrate, pH 5.5, containing 20% methanol. The gradient profile was modified according to the decrease in the content of buffer B in the mobile phase: 100% at 0.1 min, 90% at 10 min, 48% at 15 min, and 100% at 60 min. Absorbance was read at excitation and emission wavelengths of 360 and 455 nm, respectively, utilizing a Shimadzu fluorescence detector. Samples of 20 µL were used, and the concentration is expressed in µM (as the mean ± SD). Amino acids were identified by their retention time and were quantitatively determined based on their chromatographic peak area. Amino acid standards were used for calibration.

### 2.9. Statistical Analysis

The Shapiro–Wilk test was used to test the normality of the samples. Outliers were identified using the interquartile range (IQR) method and removed only when they exceeded 1.5 × IQR from the quartiles. For statistical parametric average comparison and variance comparison between groups, we performed one-way ANOVA followed by post hoc Tukey’s test or Student’s t test, as indicated, using GraphPad Prism version 5 (La Jolla, CA, USA). The data are expressed as the mean ± SEM, and significance was set at *p* < 0.05. Correlation analyses investigating the relationships between neurological scale scores and amino acid and albumin concentrations were performed within the R statistical environment (version 4.4.1; R Core Team). Pearson correlation tests (using the cor.test function) were used to evaluate the correlation between the neurological scale score and amino acid concentration. Key summary statistics—the correlation coefficient (estimate), *p* value, and confidence intervals—were extracted from the test results.

## 3. Results

### 3.1. Neurological HE Severity (Grades 0 to V) Assessed for 72 H

At 3 h post surgery, the rats showed a predominant grade I HE neurological severity, which became a predominant grade I or II at 6–12 h, and a predominant grade II at 24 h. After 30 h, the HE neurological severity became predominantly grade V and increased exponentially over time, up to 72 h (Figure 2). All the sham animals remained at grade 0 neurological severity.

**Figure 2 metabolites-15-00598-f002:**
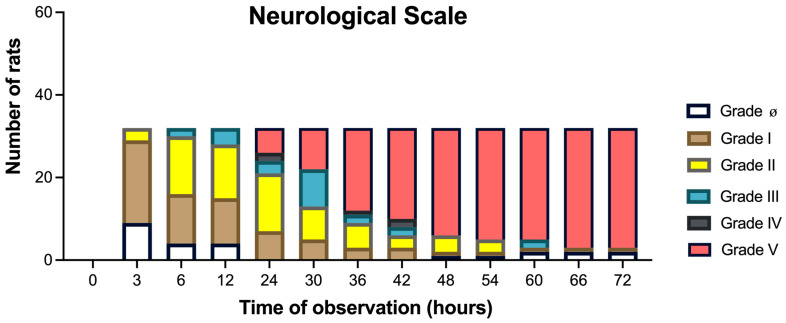
Neurological evaluation of HE severity. Neurological HE severity grades (0 to V). In hepatectomized animals, neurological parameters (including mortality rates) were evaluated every 6 h after surgery up to 72 h (n = 41). All the animals in the sham group remained at grade 0.

### 3.2. Plasma Levels of Liver Functionality Biomarkers at 24 h After Surgery

Figure 3 shows that, compared with the sham procedure, subtotal hepatectomy (Hepatec) acutely increased the blood levels of hepatic function biomarkers measured at 24 h post surgery (*p* < 0.001 for all biomarkers), indicating liver impairment. The following describes each subfigure: Figure 3A (aspartate aminotransferase: sham, 28 ± 1 U/L vs. Hepatec, 67 ± 2 U/L); Figure 3B (alanine aminotransferase: sham, 34 ± 1 U/L vs. Hepatec, 71 ± 2 U/L); Figure 3C (total bilirubin: sham, 0.04 ± 0.00 mg/dL vs. Hepatec, 1.64 ± 0.03 mg/dL); Figure 3D (direct bilirubin: sham, 0.03 ± 0.001 mg/dL vs. Hepatec, 0.54 ± 0.04 mg/dL); Figure 3E (prothrombin time: sham, 12 ± 0 s vs. Hepatec, 25 ± 1 s); and Figure 3F (ammonia: sham, 23 ± 1 µmol/L vs. Hepatec, 61 ± 2 µmol/L). All the changes in distribution of liver function markers among groups were normally distributed (Shapiro–Wilk test: Figure 3A: sham, *p* = 0.6458 vs. Hepatec, *p* = 0.6294; Figure 3B: sham, *p* = 0.5898 vs. Hepatec, *p* = 0.6104; Figure 3C: sham, *p* = 0.3632 vs. Hepatec, *p* = 0.6730; Figure 3D: sham, *p* = 0.4165 vs. Hepatec, *p* = 0.3170; Figure 3E: sham, *p* = 0.1636 vs. Hepatec, *p* = 0.2907; Figure 3F: sham, *p* = 0.3129 vs. Hepatec, *p* = 0.1983).

### 3.3. Correlations Between CSF Albumin Levels and Neurological Severity 24 h After Surgery

Figure 4 shows a correlation between increased CSF albumin levels 24 h post surgery and the adapted HE neurological severity scale (r = 0.869, *p* < 0.001). The albumin levels were as follows (grade ø–III): sham (grade ø): 1.1 ± 0.3; hepatectomy: grade I, 1.87 ± 0.26; grade II, 2.4 ± 0.4; grade III, 2.8 ± 0.3. No animals in this cohort reached grade IV at 24 h post surgery. In the CSF albumin analysis, all groups were normally distributed (Shapiro–Wilk test, grade ø, *p* = 0.3894; grade I, *p* = 0.2942; grade II, *p* = 0.8776; grade III, *p* = 0.6864).

### 3.4. Correlations of Glutamine and Amino Acid Levels in CSF with Neurological Severity Scale Scores 24 h After Surgery

Figure 5 shows that at 24 h post surgery, the increase in (i) glutamine and (ii) amino acid levels in CSF strongly correlated with the adapted HE neurological severity (grades I–IV) (*p* < 0.001 for all amino acids), except for alanine (Figure 5B, r = 0.120, *p* = 0.469). Significant correlations were observed with aspartate (Figure 5C, r = 0.790, *p* < 0.001), glutamate (Figure 5D, r = 0.853, *p* < 0.001), isoleucine (Figure 5E, r = 0.834, *p* < 0.001), leucine (Figure 5F, r = 0.813, *p* < 0.001), lysine (Figure 5G, r = 0.861, *p* < 0.001), methionine (Figure 5H, r = 0.889, *p* < 0.001), and phenylalanine (Figure 5J, r = 0.916, *p* < 0.001) from grade III and with ornithine (Figure 5I, r = 0.775, *p* < 0.001), tryptophan (Figure 5K, r = 0.814, *p* < 0.001), and valine (Figure 5L, r = 0.721, *p* < 0.001) from grade IV. In the analysis of CSF amino acids, all groups were normally distributed (Shapiro–Wilk test, Figure 5A: grade ø, *p* = 0.1373; grade I, *p* = 0.4801; grade II, *p* = 0.3370; grade III, *p* = 0.4111; grade IV, *p* = 0.9662; Figure 5B: grade ø, *p* = 0.1381; grade I, *p* = 0.1364; grade II, *p* = 0.8217; grade III, *p* = 0.1464; grade IV, *p* = 0.1880; Figure 5C: grade ø, *p* = 0.8478; grade I, *p* = 0.1197; grade II, *p* = 0.3277; grade III, *p* = 0.4330; grade IV, *p* = 0.9874; Figure 5D: grade ø, *p* = 0.9043; grade I, *p* = 0.3883; grade II, *p* = 0.7494; grade III, *p* = 0.2334; grade IV, *p* = 0.5605; Figure 5E: grade ø, *p* = 0.1601; grade I, *p* = 0.1294; grade II, *p* = 0.1104; grade III, *p* = 0.7682; grade IV, *p* = 0.7782; Figure 5F: grade ø, *p* = 0.0813; grade I, *p* = 0.9120; grade II, *p* = 0.3537; grade III, *p* = 0.1784; grade IV, *p* = 0.2429; Figure 5G: grade ø, *p* = 0.2809; grade I, *p* = 0.2267; grade II, *p* = 0.6519; grade III, *p* = 0.6841; grade IV, *p* = 0.2947; Figure 5H: grade ø, *p* = 0.0624; grade I, *p* = 0.1155; grade II, *p* = 0.7874; grade III, *p* = 0.4542; grade IV, *p* = 0.7235; Figure 5I: grade ø, *p* = 0.1105; grade I, *p* = 0.1469; grade II, *p* = 0.5739; grade III, *p* = 0.8296; grade IV, *p* = 0.0982; Figure 5J: grade ø, *p* = 0.1004; grade I, *p* = 0.1180; grade II, *p* = 0.1294; grade III, *p* = 0.5542; grade IV, *p* = 0.1455; Figure 5K: grade ø, *p* = 0.1440; grade I, *p* = 0.3937; grade II, *p* = 0.4522; grade III, *p* = 0.4903; grade IV, *p* = 0.7837; Figure 5L: grade ø, *p* = 0.1162; grade I, *p* = 0.1013; grade II, *p* = 0.2220; grade III, *p* = 0.9395; grade IV, *p* = 0.1419.

Strikingly, among the biomarkers analyzed, only CSF glutamine levels were significantly correlated with HE severity from grade I, highlighting its potential as an early biomarker of neurological dysfunction (Figure 5A, r = 0.929, *p* < 0.001).

## 4. Discussion

ALF is a life-threatening medical condition that frequently involves significant brain impairment, such as HE [45,46,47]. A reduction in hepatic workload causes an increase in plasma ammonia and in the levels of other neurotoxins that affect brain metabolism [48,49], as well as in biomarkers of liver and brain damage. In this study, we demonstrated that increased CSF levels of glutamine and albumin occur early and are positively correlated with the neurological severity of HE in an experimental model of ALF. Notably, these early increases in grade I neurological impairment underscore their potential as early biomarkers of brain dysfunction. These findings highlight the relevance of CSF levels of both glutamine and albumin, not only as indicators of HE progression but also as valuable early markers for prognostic assessment in the context of ALF.

The experimental model we used involved subtotal hepatectomy, which caused a significant increase in plasma levels of liver dysfunctionality biomarkers (AST, ALT, total bilirubin, direct bilirubin, and ammonia) and affected prothrombin time compared to sham surgery. These results reinforce those of previous studies by our group [18,35], indicating that liver dysfunction is induced by subtotal hepatectomy (Figure 3), which is consistent with clinical findings in patients with ALF [23,50,51].

The glutamatergic system plays crucial roles in cerebral homeostasis, including ammonia detoxification, primarily through glutamine production, a process crucial for brain activity [18,49]. As observed in HE, disorders in this pathway lead to significant metabolic consequences [52]. Elevated serum glutamine levels have been consistently associated with neurological decline, manifested by EEG abnormalities and poor clinical outcomes in HE patients [20]. Our findings align with our previous results, which revealed elevated levels of glutamate and glutamine in CSF in both acute HE and in vivo acute hyperammonemia models.

Our results provide further evidence that both glutamine and albumin levels in CSF are strongly correlated with neurological severity in an experimental model of ALF. Previous reports have consistently linked glutamine accumulation to the pathogenesis of hepatic encephalopathy, particularly through astrocytic metabolism and ammonia detoxification [53]. Elevated glutamine concentrations in CSF have been described as a marker of astrocytic swelling and impaired neurotransmission, which are central events in the progression of HE [54]. Our findings align with these observations and extend them by showing that glutamine is already elevated in early stages of neurological impairment (grade I), underscoring its potential as an early biomarker.

This study has some limitations that should be considered when interpreting the results. First, the sample size was relatively small, which may limit the generalizability of the findings. Furthermore, the study used only male rats, which prevents the evaluation of possible sex-related differences in response to the HE model. Finally, although we demonstrated significant correlations, the study did not include direct mechanistic analyses to demonstrate blood–brain barrier (BBB) disruption or astrocyte swelling. These points, however, pave the way for future investigations that could deepen our understanding of the underlying mechanisms. From a perspective, this study paves the way for future investigations that could deepen our understanding of the underlying mechanisms.

The positive correlation between CSF albumin levels and HE severity is a novel and intriguing observation. The albumin in serum is traditionally regarded as a marker of hepatic synthetic capacity and vascular oncotic pressure [55], and the ratio of albumin in the CSF/serum may also signal BBB disruption or increased permeability of the choroid plexus [56]. Notably, although this study did not investigate the physiopathology of HE in depth, the concomitant elevations in albumin and glutamine levels in CSF could suggest, but does not prove (because we do not analyze albumin in serum), a dual pathway contributing to HE: one reflecting structural disruption of the BBB or choroid plexus (albumin) and the other indicating astrocytic metabolic overload and edema (glutamine) [57,58]. This dual signature might offer a more comprehensive picture of the neurochemical environment in ALF and could inform the development of multimodal therapeutic strategies [17].

## 5. Conclusions

This study emphasized that the glutamine and albumin levels in CSF may serve as sensitive and early biomarkers of HE severity. Notably, these molecules could be useful biomarkers for monitoring and evaluating the progression of HE in clinical studies. Further research is warranted to explore the precise mechanisms underlying the relationship between the glutamine and albumin levels in CSF and HE progression.

## Figures and Tables

**Figure 3 metabolites-15-00598-f003:**
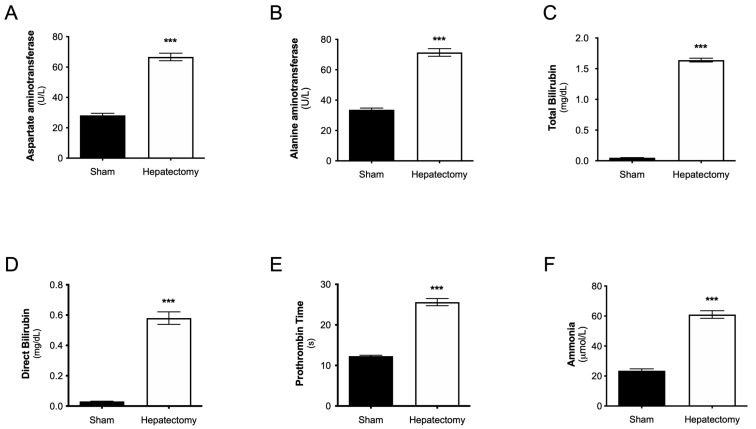
Blood levels of liver function markers 24 h after surgery. (**A**) Aspartate aminotransferase (U/L), (**B**) alanine aminotransferase (U/L), (**C**) total bilirubin (mg/dL), (**D**) direct bilirubin (mg/dL), (**E**) ammonia (µmol/L), and (**F**) prothrombin time (s). The results are expressed as the mean ± S.E.M. *** *p* < 0.001 indicates a significant difference from the sham group (Student’s t test). n: sham = 14, hepatectomy = 19.

**Figure 4 metabolites-15-00598-f004:**
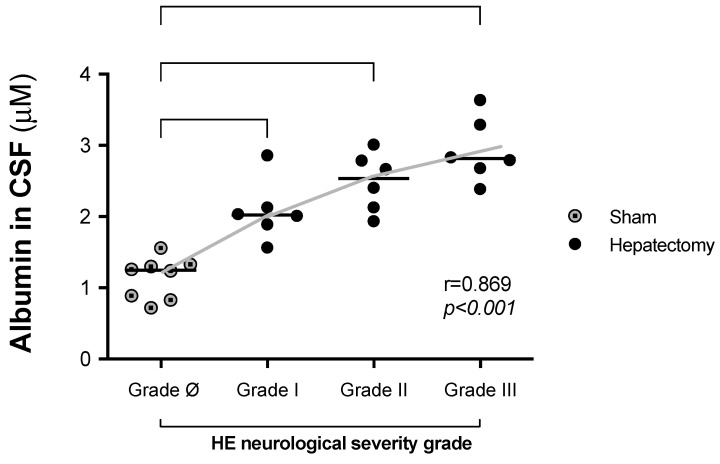
Correlation of the HE severity grade with the increase in CSF albumin levels at 24 h after surgery. CSF albumin levels: sham (grade 0) (n = 9); hepatectomy: grade I (n = 6); grade II (n = 5); grade III (n = 3). In this cohort, no hepatectomized animal remained at grade 0 or reached grade IV or V. Pearson correlation tests were performed to evaluate the correlation between the neurological scale score and the serum albumin levels (r = 0.869, *p* < 0.001).

**Figure 5 metabolites-15-00598-f005:**
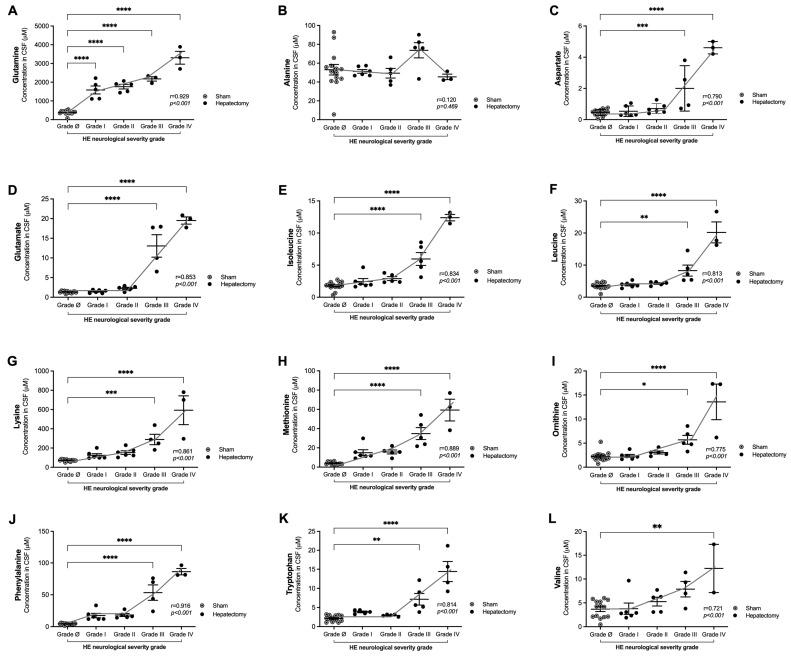
Correlation of the HE severity grade with the increase in the levels of glutamine and amino acids in CSF at 24 h after surgery. (**A**) Glutamine; (**B**) alanine; (**C**) aspartate; (**D**) glutamate; (**E**) isoleucine; (**F**) leucine; (**G**) lysine; (**H**) methionine; (**I**) ornithine; (**J**) phenylalanine; (**K**) tryptophan; and (**L**) valine. The data are expressed as the means ± SDs (µM). Sham (n = 14). Hepatectomy: grade I (n = 6), grade II (n = 6), grade III (n = 4), and grade IV (n = 3). Pearson correlation tests were performed to evaluate the correlation between the neurological scale score and amino acid concentration. One-way ANOVA followed by post hoc Tukey’s test was performed, * *p* < 0.05, ** *p* < 0.01, *** *p* < 0.001, and **** *p* < 0.0001.

**Table 1 metabolites-15-00598-t001:** The adapted neurological HE severity grades (0 to V).

Neurological Grades	Hepatic Encephalopathy Neurological Scale for Rats
ø	Freely moving and exploring the home cage. No reflex alterations.
I	No spontaneous exploratory activity. Righting reflex present. Researcher is not able to place the animal on its back. Corneal reflex preserved.
II	No spontaneous exploratory activity. Researcher is not able to place the animal on its back, but the animal is able to return to normal position within 3 s. Corneal reflex preserved.
III	No spontaneous exploratory activity. Loss of righting reflex–animal is not able to return to normal position. Corneal reflex preserved.
IV	No spontaneous exploratory activity or locomotion. Loss of reflexes and no response to pain stimuli.
V	Death

## Data Availability

The data presented in this study are available from the corresponding author upon request.

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
