# Peer review of "Glutamine and Albumin Levels in Cerebrospinal Fluid Are Correlated with Neurological Severity in an Experimental Model of Acute Hepatic Encephalopathy"

_metabolites, 2025, doi:10.3390/metabo15090598_

Round 1

Reviewer 1 Report

Comments and Suggestions for Authors

The manuscript by Pedro Arend Guazzelli and colleagues is interesting.
Some suggestions:

1. Include the full name in the title and not the abbreviation (CSF).
2. There are five figures in the methodology section. Is this correct? Or are they results?
3. Was it a subtotal hepatectomy or a partial hepatectomy? If subtotal, what was the mortality rate?
4. Provide several references to explain the surgical procedure. Which one did you use? Did you use reports from five authors?
5. Given the type of study, I suggest using ANCOVA for the statistics.
6. I strongly suggest not mixing methodology and results. Please include figures and tables where appropriate.
7. The amount of information is very extensive, but the discussion is very short. Could you expand on it?
8. What are the perspectives and limitations of the study?
9. There are very old references, from 1994, I suggest updating the references.

Author Response

 Reviewer 1

Comments: The manuscript by Pedro Arend Guazzelli and colleagues is interesting.
Some suggestions:

Response: Thank you for your thoughtful review and suggestions, which have certainly helped us improve our manuscript.

Comment 1: Include the full name in the title and not the abbreviation (CSF).

Response: Thank you for your suggestion. We have changed the manuscript title to include the full name of cerebrospinal fluid, as requested. The change can be seen on page 1, in the article title.

Comment 2: There are five figures in the methodology section. Is this correct? Or are they results?

Response: Thank you for the observation. There was a formatting error after manuscript submission. Please accept our apologies. Figures 2 through 5 have been moved to the Results section, as they represent the study's results, not its methodology. This relocation can be found starting on page 10, in the Results section.

Comment 3: Was it a subtotal hepatectomy or a partial hepatectomy? If subtotal, what was the mortality rate?

Response: Thank you for your comment. We have clarified in the Methodology section that the hepatectomy performed was subtotal. We have added the post-surgical mortality rate, which was 90.6%, according to the study data (Figure 2), as detailed on page 5 in the Methodology section.

Comment 4: Provide several references to explain the surgical procedure. Which one did you use? Did you use reports from five authors?

Response: Thank you for your comment. The protocol was based on Guazzeli et al. 2019, and details can be found in the References section (Reference 35). The explanation of the procedure has been revised on page 5 in the Methodology section.

Comment 5: Given the type of study, I suggest using ANCOVA for the statistics.

Response: We thank the reviewer for this valuable suggestion. In accordance with the recommendation, we re-analyzed the data using ANCOVA. The results obtained were consistent with those reported in our original analysis, without altering the overall conclusions. Respectfully, for clarity and readability, we would prefer to maintain the original presentation of the results in the manuscript.

Method description: Analysis of covariance (ANCOVA) was employed to investigate the effects of Group and Scale on each amino acid's concentration. The data were log10 transformed to stabilize variance and normalize distribution. FDR adjustment of p-values was performed to control for multiple comparisons. The results were visualized using a heatmap to highlight significant effects across different amino acids.

Comment 6: I strongly suggest not mixing methodology and results. Please include figures and tables where appropriate.

Response: Thank you for your observation. There was a formatting error after manuscript submission. Please accept our apologies. We have reorganized the manuscript to clearly separate the Methodology and Results sections. All figures and tables were reordered and positioned closer to the first citation in the text for clarity. The changes can be seen throughout the manuscript.

Comment 7: The amount of information is very extensive, but the discussion is very short. Could you expand on it?

Response: We thank the reviewer for this important observation. In the revised version of the manuscript, we have substantially expanded the Discussion section (pages 15 and 16) to better contextualize our findings within the current literature. Specifically, we elaborated on the role of CSF glutamine and amino acids as markers of astrocytic dysfunction and neurotransmission impairment, and integrated these results with recent studies on ammonia detoxification and brain edema in hepatic encephalopathy. Furthermore, we discussed the potential clinical implications of our findings for early detection and monitoring of neurological deterioration in acute liver failure. We believe these additions provide a more comprehensive interpretation of the data and strengthen the overall impact of the study.

Comment 8: What are the perspectives and limitations of the study?

Response: Thank you for the comment. We have added a detailed section on the limitations of our study and prospects for future research. The new section can be found on pages 14–15.

Comment 9: There are very old references, from 1994, I suggest updating the references.

Response: Thank you for your comment. We have revised the reference list and replaced older references with more recent and relevant studies. The updated reference list can be found at the end of the manuscript.

Reviewer 2 Report

Comments and Suggestions for Authors
  1. The paper lacks information on outlier handling, normality testing (e.g., Shapiro-Wilk), and post-hoc tests used after ANOVA. This makes it difficult to assess data robustness.
  2. The methods do not describe how animals were assigned to groups (randomization) or whether neurological assessments were performed blindly to avoid observer bias.

  3. Critical details are missing, such as survival rates after 92% hepatectomy, duration of liver ischemia during vessel ligation, or specific exclusion criteria beyond the three deaths mentioned.

  4. For amino acid and albumin HPLC analysis, key validation parameters are omitted: LOD/LOQ (limits of detection/quantification), recovery rates of standards, and intra-/inter-day precision.

  5. While correlations with HE severity are shown, the paper fails to discuss how glutamine and albumin could be applied clinically (e.g., for early diagnosis or monitoring). No comparison is made with established biomarkers (e.g., ammonia).

  6. The claim that albumin reflects BBB disruption is unsupported by histology/immunochemistry (e.g., no GFAP staining for astrocyte swelling). This weakens mechanistic conclusions.

  7. Graphs (A–L) lack units on the Y-axis (presumably μM) and statistical significance markers between individual groups (e.g., asterisks for grade I vs. sham comparisons).

Recommendations for improvement:

1. Add a "Study Limitations" section (small sample size, exclusion of female rats, lack of direct mechanistic proof).

2. Clarify how anesthesia effects (isoflurane) were controlled in neurological assessments.

3. Include serum glutamine/albumin levels alongside CSF data to evaluate biomarker specificity.

Author Response

Reviewer 2

Comment 1: The paper lacks information on outlier handling, normality testing (e.g., Shapiro-Wilk), and post-hoc tests used after ANOVA. This makes it difficult to assess data robustness.

Response: We appreciate the reviewer’s observation regarding the statistical analysis. In the revised version of the manuscript, we have added detailed information about data processing and statistical testing. Outliers were identified using the interquartile range (IQR) method and removed only when they exceeded 1.5 × IQR from the quartiles. Normality of the data was assessed with the Shapiro–Wilk test. For comparisons between multiple groups, one-way ANOVA was applied when assumptions were met. These details have now been included in the Methods section (page 9) and Results section (pages 11, 12 and 14) to enhance the transparency and robustness of the statistical analysis.

Comment 2: The methods do not describe how animals were assigned to groups (randomization) or whether neurological assessments were performed blindly to avoid observer bias.

Response: Thank you for your observation. We agree that transparency in describing the methodology is crucial. We have clarified the following in the revised version of the manuscript: "all surgical procedures were performed randomly" (page 5) and "neurological evaluations were conducted blindly" (page 6). These changes can be found in the Methodology section.

Comment 3: Critical details are missing, such as survival rates after 92% hepatectomy, duration of liver ischemia during vessel ligation, or specific exclusion criteria beyond the three deaths mentioned.

Response: Thank you for your comment. We have clarified in the Methodology section that the hepatectomy performed was subtotal. We have added the post-surgical mortality rate, which was 90.6%, according to the study data (Figure 2); details are provided on page 5 in the Methodology section.

Comment 4: For amino acid and albumin HPLC analysis, key validation parameters are omitted: LOD/LOQ (limits of detection/quantification), recovery rates of standards, and intra-/inter-day precision.

Response: We thank the reviewer for pointing out the need to provide validation parameters for the analysis of amino acids by HPLC. These parameters are consistent with previously reported methods for amino acids and small peptide HPLC analysis (Joseph & Marsden, 1986). We have included the following corresponding details in the Methods section (Section 2.6, page 7):

“The evaluation of amino acids CSF concentration was performed through HPLC in partnership with Laboratory Genetics Unit, Medical Genetics Service, Clinicas Hospital of Porto Alegre (HCPA, RS, Brazil), when it was evaluated: i) Limit of detection/quantification (LOD/LOQ - 2.0 μM); ii) Precision: intra-day (6.0%), inter-day (8.5%); and iii) Recovery of standards (ranged between 82% and 120%), accordingly to (Joseph et al.).”

Comment 5: While correlations with HE severity are shown, the paper fails to discuss how glutamine and albumin could be applied clinically (e.g., for early diagnosis or monitoring). No comparison is made with established biomarkers (e.g., ammonia).

Response: Accordingly, we have included a paragraph about the limitations and perspectives of our study.

From a clinical perspective, our results highlight the translational potential of using CSF glutamine and albumin as biomarkers for monitoring HE severity. Compared with ammonia, the most widely studied biomarker, glutamine and albumin may provide complementary information. Ammonia levels often lack sensitivity and specificity, as they do not always correlate with neurological status. In contrast, glutamine elevation directly reflects cerebral ammonia detoxification, while albumin may signal barrier disruption. This dual signature could improve the prognostic accuracy of biomarker panels in patients with ALF and HE.

Comment 6: The claim that albumin reflects BBB disruption is unsupported by histology/immunochemistry (e.g., no GFAP staining for astrocyte swelling). This weakens mechanistic conclusions.

Response: We agree with the reviewer that the lack of histological or immunohistochemical confirmation is a limitation of the present study. We have mitigated our conclusions accordingly, clarifying in the Discussion (page 16) that the increase in CSF albumin suggests—but does not prove—BBB disruption. We now explicitly state that further mechanistic studies with histological and immunohistochemical analyses (e.g., GFAP staining) are warranted to confirm this hypothesis.

Comment 7: Graphs (A–L) lack units on the Y-axis (presumably μM) and statistical significance markers between individual groups (e.g., asterisks for grade I vs. sham comparisons).

Response: Thank you for noting this. We have revised Figures 3–5 to include proper units (μM) on all Y-axes. Additionally, we have added statistical significance markers (asterisks) to indicate pairwise differences between sham and HE (grades I–IV). These changes improve the clarity and interpretability of the figures.

Recommendations for improvement:

Comment 8: Add a "Study Limitations" section (small sample size, exclusion of female rats, lack of direct mechanistic proof).

Response: We appreciate this insightful suggestion. Unfortunately, due to sample availability, serum glutamine and albumin were not measured in this study. We recognize this as a limitation and have explicitly stated it in the Discussion (page 15). Importantly, we emphasize that future studies should include both CSF and serum measurements to determine biomarker specificity and to better evaluate their translational potential for clinical application.

“This study has some limitations that should be considered when interpreting the results. First, the sample size was relatively small, which may limit the generalizability of the findings. Furthermore, the study used only male rats, which prevents the evaluation of possible sex-related differences in response to the HE model. Finally, although we demonstrated significant correlations, the study did not include direct mechanistic analyses to demonstrate blood-brain barrier (BBB) ​​disruption or astrocyte swelling. These points, however, pave the way for future investigations that could deepen our understanding of the underlying mechanisms. From a perspective, this study paves the way for future investigations that could deepen our understanding of the underlying mechanisms.”

Comment 9: Clarify how anesthesia effects (isoflurane) were controlled in neurological assessments.

Response: We appreciate the reviewer's observation. We would like to clarify that all neurological assessments were performed only after the animals had fully recovered from anesthesia, to avoid any interference of isoflurane on the results. Furthermore, all experiments used animals from the sham group, which underwent the same surgical procedure and received isoflurane anesthesia but did not undergo hepatectomy. We observed that none of the animals in the sham group showed any changes in their neurological scale scores, which reinforces that the anesthesia factor did not influence the results obtained. This information has been added to the Methodology section (page [page number]) for greater clarity.

Comment 10: Include serum glutamine/albumin levels alongside CSF data to evaluate biomarker specificity.

Response: Thank you for your comment. As specified in the response to comment 8, this topic is a limitation of our study.

Round 2

Reviewer 1 Report

Comments and Suggestions for Authors

congratulations

Reviewer 2 Report

Comments and Suggestions for Authors

I have reviewed the revised version of your manuscript and would like to commend you for thoroughly addressing all the comments and concerns raised during the peer-review process. Your detailed responses and the subsequent revisions have significantly improved the quality and clarity of the manuscript. The additional data, clarifications, and refined interpretations have strengthened your results and discussion, making the paper more robust and compelling.